# Twenty-Month Monitoring of Humoral Immune Response to BNT162b2 Vaccine: Antibody Kinetics, Breakthrough Infections, and Adverse Effects

**DOI:** 10.3390/vaccines11101578

**Published:** 2023-10-10

**Authors:** Jaroslaw Walory, Iza Ksiazek, Michal Karynski, Anna Baraniak

**Affiliations:** 1Department of Biomedical Research, National Medicines Institute, 00-725 Warsaw, Poland; 2Department of Biochemistry and Biopharmaceuticals, National Medicines Institute, 00-725 Warsaw, Poland; i.ksiazek@nil.gov.pl; 3Department of Falsified Medicines and Medical Devices, National Medicines Institute, 00-725 Warsaw, Poland; m.karynski@nil.gov.pl

**Keywords:** anti-SARS-CoV-2 antibody kinetics, breakthrough infections, COVID-19, mRNA vaccine, vaccine safety assessment

## Abstract

Background: Vaccination is one of the most effective life-saving medical interventions, and the introduction of SARS-CoV-2 vaccines was intended to prevent the serious implications of COVID-19. The objectives of the study were (i) to observe the humoral immune response to the BNT162b2 vaccine and SARS-CoV-2 infection (mainly breakthrough infections), (ii) to demonstrate the persistence of anti-SARS-CoV-2 antibodies over time in relation to the number of received vaccine doses and the course of infection, and (iii) to determine the adverse effects after primary vaccine doses. Methods: To assess the humoral response, IgG and IgA anti-S1 antibodies were quantified by ELISA assays. In total, the tests were carried out seven times in almost two years. Results: We demonstrated strong immunogenicity (compared to levels before primary vaccination, 150- and 20-fold increases in IgG and IgA, respectively) of the BNT162b2 vaccine. Over time, we observed a systematic decline in antibody levels, which may have contributed to breakthrough infections. Although they caused seroconversion similar to the booster, antibody levels in such patients fell more rapidly than after re-vaccination. On the other hand, in individuals who did not receive booster(s) and who did not present breakthrough infection, anti-SARS-CoV-2 antibodies returned to pre-vaccination levels after 20 months. The most commonly recognized adverse effects were injection site redness and swelling. Conclusion: Vaccination is highly effective in preventing the most severe outcomes of COVID-19 and should be performed regardless of prior infection. Booster doses significantly enhance anti-SARS-CoV-2 antibody levels and, in contrast to those obtained by breakthrough infection, they remain longer.

## 1. Introduction

The global spread of severe acute respiratory syndrome coronavirus 2 (SARS-CoV-2) causing coronavirus disease 2019 (COVID-19) resulted in pandemic status for the illness by the World Health Organisation (WHO) [1]. Since the first identification of SARS-CoV-2 in December 2019, the WHO has recorded more than 770 million confirmed cases of COVID-19, including almost 7 million deaths [2]. Vaccination is one of the most effective and cost-efficient public health interventions to prevent infectious diseases [3]. There are several COVID-19 vaccines validated for use by the WHO. The mass vaccination programme started in early December 2020, and the number of vaccine doses currently administered worldwide is more than 13 billion [2,4].

The first COVID-19 vaccine recommended by the WHO was the BNT162b2 (Comirnaty^®^, Pfizer, Philadelphia, PA, USA and BioNTech, Mainz, Germany), which consists of a nucleoside-modified mRNA encoding spike (S) protein, specific to the Wuhan-Hu-1 strain isolated in China during the first outbreak in late 2019, formulated in lipid nanoparticles [4,5]. Transient expression of the S antigen induces neutralising antibodies and cellular immune responses providing defence from COVID-19 [5]. Data obtained from clinical trials have shown that a two-dose scheme of the vaccine, administered 21 days apart, offered 86% and 95% protection against infection and severe disease, respectively [4,6,7]. The emergence of SARS-CoV-2 mutants, carrying changes mainly in the S protein, led to increased transmission and/or infectivity of the virus and reduced vaccine efficacy, resulting in infections of vaccinated individuals [8,9,10,11]. In addition, several studies have shown that the antibody levels decline markedly in six months following primary vaccination, which may also contribute to an increase in breakthrough infections [12,13,14,15,16]. Booster doses were aimed at enhancing the immune response to provide long-term protection against COVID-19, including that caused by the variants of concern (VOCs) [6,7,16,17]. For Alpha, Beta, Gamma, and Delta variants, the effectiveness of the vaccine in preventing infection and severe disease remained similar as the assumed efficacy of the ancestral strain, while for Omicron sub-lineages (including BA.1, BA.2, BA.5) it amounts to 44% and 72%, respectively [7]. Furthermore, vaccination after recovery from SARS-CoV-2 infection resulting in a ‘hybrid’ immunity was found to significantly increase the strength of the humoral response [7,18,19,20,21].

Vaccine effectiveness, in contrast to vaccine efficacy assessed in clinical trials, is based on a reduction in the risk of infection/disease among vaccinated individuals in real life. This can be influenced by many factors, including internal host factors (e.g., age, gender, genetics, co-morbidities), external host factors (e.g., pre-existing immunity, microbiota, past infections), environmental factors (e.g., geographical location, season, family size) and behavioural factors (e.g., smoking, alcohol consumption, exercise, rest/sleep). In addition, factors related to the kind of vaccine (product type, adjuvant, dose) and factors associated to administration (schedule, site, route, time of vaccination, concurrent other vaccines and medicines) are also important [22,23]. Despite the high vaccine effectiveness, some people have concerns about receiving the COVID-19 vaccine related to its safety and side effects [24,25,26]. Therefore, a better understanding of adverse reactions of individual vaccines allows for more informed decisions about vaccination.

The VAERS and EudraVigilance vaccine safety systems (maintained by the Food and Drug Administration and the European Medicines Agency, respectively) were established to report vaccine adverse reactions, including monitoring the safety profile of COVID-19 vaccines administered to the public [27,28]. The most commonly recognised side effects within seven days of each vaccine dose consisted of injection site redness and swelling, headache, myalgia, and fatigue. Severe reactions with facial paralysis, anaphylaxis, and cerebral venous sinus thrombosis were rare and occurred at similar rates in vaccinated and unvaccinated individuals [24,29]. Most research on vaccine side effects has focused on clinical trials or pre- and post-intervention investigations. Over time, more observational studies have emerged, and these have provided data on the safety of COVID-19 vaccine in real-world settings [26,30].

The aim of the study was a long-term monitoring (total of seven checkpoints in almost two years) of the humoral immune response to BNT162b2 vaccine (two primary doses and two boosters) and SARS-CoV-2 infection (including mainly breakthrough infections) of a well-characterized relatively invariable patient group. In addition, adverse effects reported after the primary vaccine doses were defined.

## 2. Materials and Methods

### 2.1. Study Design, Data Collection, and Cohort Characteristics 

The investigation was designed to monitor the humoral response in IgG and IgA classes of anti-SARS-CoV-2 antibodies. It was carried out as part of routine diagnostics in the medical laboratory of the National Medicines Institute (NMI) and was attended by NMI workers. The qualified number of subjects (*n* = 91) is representative of the adult Polish population (30 million), assuming the confidence of the results obtained α = 0.95 and the maximum error of the result up to 10%.

The estimate of the maximum error in the representativeness of the results for the population of the country was calculated from the formula:$$N_{min}=\frac{N_{P}\left(\alpha^{2}\cdot f\left(1-f\right)\right)}{N_{P}\cdot e^{2}+\alpha^{2}\cdot f\left(1-f\right)}$$
where: 

*N_min_*—minimal sample size;*N_p_*—size of the population from which the sample is collected;α—confidence level for the results, value of the Z result in a normal distribution for the assumed level of significance 1.96;f—fraction size;e—maximum error, expressed as a fractional number, e.g., 10% is 0.1.

All study participants received primary vaccination of BNT162b2 and some also were administered booster dose(s) of this vaccine. The timeline of vaccine administrations and blood donation checkpoints, as well as the conducted questionnaires, is shown in Figure 1.

Information collected from the questionnaires consisted of patient characteristics (age, gender, body weight/growth, and chronic diseases), a history of SARS-CoV-2 infection, and the occurrence of vaccine side effects. Data analysis including patient characteristics and antibody kinetics was performed for the total cohort and by type of acquired immunity (‘vaccine only’ or ‘hybrid’), as well as by age above/below 50 years. The correlation of seroconversions with the presence of chronic disease or body mass index (BMI) was also investigated.

### 2.2. Serum Collection and Anti-SARS-CoV-2 Antibody Tests

Blood donations were performed according to the timeline shown in Figure 1. Serum samples, obtained by whole blood centrifugation, were stored at −20 °C pending antibody testing. To assess the humoral response, IgG and IgA antibodies were quantified by enzyme linked immunosorbent assay (ELISA) using an Infinite^®^ M1000 PRO instrument (Tecan Trading AG, Männedor, Switzerland). The IgG and IgA antibodies against S protein (corresponding to IgG-S1 and IgA-S1) and IgG antibodies against nucleocapsid (N) protein (IgG-NCP) were tested with three commercial tests, Anti-SARS-CoV-2 QuantiVac-ELISA (IgG), Anti-SARS-CoV-2 ELISA (IgA), and Anti-SARS-CoV-2-NCP ELISA (IgG) (EUROIMMUN Medizinische Labordiagnostika AG, Lübeck, Germany), respectively. The obtained results were interpreted according to the manufacturer’s guidelines. Due to the low levels of IgA-S-1, their determination was waived at the seventh checkpoint.

### 2.3. Statistical Analysis

The results were analysed with GraphPad Prism v 8.0.1 (GraphPad Software, San Diego, CA, USA). Various assays, including the two-tailed Wilcoxon matched-pairs signed rank test, Mann–Whitney test or Chi-square test, were used in calculating the *p*-value at α < 0.05.

## 3. Results

### 3.1. Participant Characteristics

A total of 91 participants were included in the study and their characteristics are shown in Table 1. Until the third blood donation, the number of patients remained unchanged. However, for the kinetic analyses, the following checkpoints were represented by 90, 86, 85, and 76 subjects, respectively.

### 3.2. Antibody Levels

The kinetics of IgG-S1 and IgA-S1 antibodies, for the total cohort and by type of immunity, as well as by age above/below 50 years, are shown in Table 2. Meanwhile, IgG-NCP levels are provided in Appendix A. To track trends in IgG-S1 and IgA-S1 antibody levels after primary vaccine doses in intact systems (with vaccine-induced immunity only), three individuals who contracted SARS-CoV-2 infection between the second vaccine dose and the second blood donations were excluded from the analysis. In addition, one of these subjects finished participation in the study after the fourth checkpoint. Antibody levels for the sixth and seventh blood donations include all participants tested at these checkpoints, except those previously ruled out.

Joining the study, 26 individuals possessed a recovered status (positive results for IgG-NCP in the first blood donation; Appendix A). They were found to have higher antibody levels for IgG-S1 and IgA-S1 classes (seven- and ten-fold, respectively) than those without prior infection. Nevertheless, antibody levels at the second checkpoint for the total cohort were only slightly higher with ‘hybrid’ immunity than with ‘vaccine only’ status (*p* = 0.6659). Strong immunogenicity of the vaccine (increase in IgG-S1 antibody titres by almost 150-fold and IgA-S1 by more than 20-fold) was observed in all study cohorts. Despite the differences in mean levels of antibody, they were not statistically significant in individuals under 50 years old in the ‘vaccine only’ and ‘hybrid’ immunity groups. In contrast, a statistically significant difference (*p* = 0.0154) in antibody levels between groups with different immunization statuses was observed in participants over 50 years old.

At the following checkpoints (up to the fifth blood donation), a systematic decrease in antibody levels from baseline values obtained at the second checkpoint was observed in all studied groups (Figure 2 and Figure 3). The decrease in antibodies of individuals with ‘vaccine only’ status was higher than in those with ‘hybrid’ immunity by about 13% and 30% in IgG-S1 and IgA-S1, respectively. At the fifth checkpoint, the IgG-S1 and IgA-S1 antibody levels decreased by about 90% and 80%, respectively. Between the fifth and sixth checkpoints, 63 participants received the third dose of the vaccine. Despite the fact that 20 individuals did not accept the first booster, the average levels of antibody at the sixth blood donation increased compared to the fifth checkpoint in all studied groups. The growing trend in antibody levels remained until the seventh blood donation, with eight subjects receiving the second booster before this checkpoint.

No correlation was observed between chronic disease or BMI of participants and the seroconversions or duration of antibody persistence.

### 3.3. Breakthrough Infection

All SARS-CoV-2 infections were confirmed by RT-qPCRs or rapid antigen tests (RATs) using the assays that we described previously [31,32]. Immune breakthrough levels were determined as IgG-S1 and IgA-S1 antibody levels obtained after vaccination (primary doses and boosters) and prior to positive IgG-NCP antibody levels. The results of antibody breakthrough levels, for the total cohort and by age of individuals within type of immunity, are shown in Table 3. Meanwhile, IgG-NCP levels are provided in Appendix A. Finally, all participants who were positive for IgG-NCPs had previously identified infections by RT-qPCRs or RATs. As infections occurred in the interval between checkpoints, the real breakthrough levels were the same or less than those assigned. In the ‘hybrid’ immune status group, breakthrough levels in the IgG-S1 and IgA-S1 classes were identified for individuals with positive results and by increasing IgG-NCP levels after vaccination relative to previously established IgG-NCP levels. Consecutive positive results for IgG-NCP characterized by a decreasing trend were not included in the analysis and they were considered single infections.

After primary vaccination, 33 individuals (36.2% of all participants) presented with breakthrough infections (Appendix A). Two subjects were infected twice, one three times. In seven cases (19.4% of breakthrough infections), they had full-blown course of the disease with 1–3 days of fever, muscle aches, sore throat, cough, or gastrointestinal symptoms. The others mostly complained of a sore throat and/or headache. Overall, breakthrough infections were found four times more often in individuals with ‘vaccine only’ status than with ‘hybrid’ immunity. They were also more common in those under 50 years old (*n* = 21; 58.3%) than in the elderly (*n* = 15; 41.7%).

The first three breakthrough infections occurred in just under four months after receiving the primary vaccine doses. Despite an increase in average antibody levels after infection (1.4- and 3.5-fold for IgG-S1 and IgA-S1, respectively), there was no significant immune enhancement (*p* > 0.05). All patients with breakthrough infections between the fifth and seventh checkpoints received a booster vaccine dose, so the increases in antibody levels at the sixth and seventh blood donations are the results of both the booster and the infection. Average antibody levels at the sixth checkpoint were significantly higher than those observed at the fifth by 11.5 and 10.4 times for IgG-S1 and IgA-S1, respectively (*p* = 0.0001). Also at the last sampling, the mean level of IgG-S1 antibodies increased compared to the previous checkpoint (*p* = 0.0017).

### 3.4. Persistence of Anti-SARS-CoV-2 Antibodies over Time

The temporal correlation of anti-SARS-CoV-2 antibody levels with the number of received vaccine doses in recovered (excluded infection before primary vaccination) and uninfected individuals is shown in Table 4. However, it should be noted that some groups were represented by only a few participants.

Twenty months after primary vaccination, an almost complete decrease in both IgG-S1 and IgA-S1 was observed in individuals who remained uninfected and did not receive any reminder dose. In contrast, adoption of the booster(s) by such subjects increased antibody levels by around eight-fold in both classes. On the other hand, although infection initially increased antibody levels (20 and 11 times for IgG-S1 and for IgA-S1, respectively), they were found to decrease by as many as half at the next checkpoint. As in uninfected individuals, the admission of booster dose(s) by recovered patients resulted in increase in both classes of antibodies.

### 3.5. Vaccine Safety Assessment

The reported side effects of vaccination (SEV) included injection site pain (ISP), injection site swelling (ISS), injection site redness (ISR), pain/enlargement of lymph nodes (PLN), fatigue (F), headache (H), myalgia (M), chills (Ch), arthralgia (A), temperature over 38 °C (T > 38 °C), nausea or vomiting (N/V), and hypersensitivity reaction (HR). The prevalence of SEV with mean duration and declaration of medication admission (MA) for the total cohort and by immune status (including the age range of study participants), separately for the first and second vaccine doses, is shown in Figure 4 and Figure 5 and Table 5.

SEV occurred after both doses of vaccination in all participants. Except for injection site pain and injection site swelling, the vast majority of adverse reactions for the total cohort were more frequently observed after the second dose of vaccination than the first. Moreover, medication admission (antipyretics only) was also more often reported in this period.

The most common symptoms after the first vaccine dose were injection site pain, followed by fatigue and injection site swelling. However, as many as half of those with ‘hybrid’ immunity suffered from fatigue, while in the group of individuals with ‘vaccine only’ status, only every third. In addition, in both these populations, this symptom occurred more frequently in patients under 50 years old. As previously, after the second dose of vaccination, the most common SEV for the total cohort were injection site pain and fatigue. However, after this dose, the next most frequent adverse reaction was headache. Both fatigue and headache were most prevalent in individuals under 50 years old with ‘vaccine only’ immunity than in the elderly or those with ‘hybrid’ status.

## 4. Discussion

Although there are no recommendations for routine testing of anti-SARS-CoV-2 antibody levels, monitoring the immune response after vaccination and/or illness is one of the key steps in studies of vaccine efficacy and/or population immunity. The benefits of such investigations include the assessment of baseline seroprevalence of SARS-CoV-2 infection in unvaccinated individuals, early identification of poor or non-responders to vaccination, and timely detection of more rapid decline in anti-SARS-CoV-2 antibody levels. Such knowledge is crucial for making rational decisions about booster doses and managing a possible next wave of the COVID-19 pandemic [33].

In Poland, the mass vaccination programme started on 27 December 2020 and, in its initial phase, targeted health care workers (HCWs), most of whom received two primary doses by the end of March 2021 [34]. In this study, a group of 91 HCWs vaccinated with BNT162b2 were followed up for almost two years considering antibody levels before vaccination (baseline seroprevalence determination), after primary vaccination (between four and six checkpoints for participants without or with a booster, respectively), and after booster dose(s) (up to two checkpoints).

In contrast to previous reports from Poland presenting a very low (up to 2.4%) seroprevalence in HCWs, the current investigation demonstrated recovery status in 28.6% of the tested participants [35]. This difference is most likely due to the fact that most of these surveys were conducted in May–August 2020 with relatively fewer SARS-CoV-2 infections in Poland at that time compared to other European countries, whereas in our study, the first testing of anti-SARS-CoV-2 antibody levels occurred in January 2021, just after the peak incidence of COVID-19 in the country in October–November 2020 [2,8,36]. The timing of the first examination, scheduled up to two days before the first vaccine dose, allowed comparison of the immunogenicity induced by the infection with that of the vaccine. Vaccination was found to be more immunogenic (around 40 and 5 times in IgG and IgA antibody classes, respectively) than illness which is consistent with earlier reports [20,35,37,38]. Over time, a systematic decrease in antibody levels was observed compared to baseline values obtained after the primary vaccination. The decrease in antibodies of individuals with ‘vaccine only’ status was higher than with ‘hybrid’ immunity. Our observations are in line with studies from other groups, so it seems that full vaccination of convalescents is completely justified, and that vaccination of people without previous COVID-19 disease with booster doses is even necessary [20,37,39]. In contrast to the susceptibility to SARS-CoV-2 infection, no correlation was found between the presence of chronic diseases or the BMI value of study participants with the induction and duration of antibody persistence. The immune response in patients with chronic diseases depends on the type of co-morbidity and may remain the same as in healthy individuals, or it may be retarded or severely disabled [40,41]. On the other hand, conclusions regarding the correlation of BMI and neutralising antibody titres vary between reports [12,42]. Therefore, considering that vaccination primarily prevents the severe outcomes of COVID-19, it seems that it should be performed regardless of patient characteristics, unless there are obvious obstacles to it. Overall, after eight months, antibody levels in study participants decreased by about 90%, only to fall back to pre-vaccination levels after 20 months in those who did not receive any booster. As the antibody response gradually weakens after the primary vaccination, this contributes to an increase in breakthrough infections. SARS-CoV-2 infections were found more often in individuals with ‘vaccine only’ status than with ‘hybrid’ immunity, which confirms the findings of other researchers that the humoral response and vaccine effectiveness in previously infected patients is better than in those receiving only vaccine alone [12,20,21,35,37]. Although the breakthrough infections in our study resulted in seroconversion similar to the booster, at the last checkpoint the antibody levels of those who recovered (and did not accept booster) were almost half those of participants who received a subsequent vaccine dose(s). This observation once again confirms the legitimacy of accepting a booster(s), despite the breakthrough infection [18,19,20,21].

Due to the emergence of VOCs and differences in the neutralisation levels of mutants by antibodies produced by the vaccine based on the Wuhan-Hu-1 strain, it is not possible to estimate a common level of antibodies that would determine the humoral protective response. In our study, the first patient with an Alpha variant breakthrough infection was identified just two weeks after the second baseline vaccine dose, as we previously reported [10]. Although the expected efficacy of BNT162b2 against this variant was similar to that of the wild-type strain, in this case it proved to be unsatisfactory [7]. Unfortunately, the level of anti-SARS-CoV-2 antibodies was not tested in this individual after the second dose of vaccine, immediately before the infection. Therefore, we could not determine whether insufficient antibodies produced after vaccination contributed to the disease. However, it should be noted that despite co-morbidities that would have put the patient at risk of an unfavourable prognosis of infection, the participant presented only mild cold symptoms. Only two other individuals developed SARS-CoV-2 infections less than four months after receiving baseline vaccination doses. The other breakthrough infections were identified at a time when there were already significant declines in antibody levels. These infections were most likely due to the Omicron variant, which emerged in Poland in December 2021 [8].

The most commonly recognized SEVs of each vaccine dose were redness and swelling at the injection site, as well as fatigue and headache. Interestingly, SEVs after the first dose of vaccination were more often reported by participants with ‘hybrid’ immunity than those who had not been previously infected. The opposite was found for SEVs after the second dose of vaccination, where they were more often presented by those with a ‘vaccine only’ status than by recovered individuals. Probably, the observed correlations resulted from a more violent reaction of the patients’ immune systems during re-exposure (second contact) to the S protein of virus. In addition, it was found that those who had more frequent SEVs were also more likely to take antipyretics.

Our study was conducted on a relatively small number of participants; however, it is representative of the adult population of the country, with a maximum error of the results up to 10%. As the study group consisted of four times as many women as men, we did not perform a gender analysis. An important aspect of our study was to demonstrate the persistence of anti-SARS-CoV-2 antibodies over time in a relatively unchanged group of patients depending on the number of vaccine doses received and the course of infection.

## 5. Conclusions

We demonstrated the strong immunogenicity of the BNT162b2 vaccine in both antibody classes, IgG and IgA. Over time, a systematic decrease in antibody levels was observed compared to baseline values obtained after primary vaccination. Booster doses significantly enhance anti-SARS-CoV-2 antibody levels and, in contrast to those obtained by breakthrough infection, they remain longer. Although neutralising antibodies are not the only factor responsible for protection against the morbidity and course of COVID-19, they are a good marker to assess the immune status of patients, the effectiveness of the vaccine, and the epidemic situation of the population.

## Figures and Tables

**Figure 1 vaccines-11-01578-f001:**
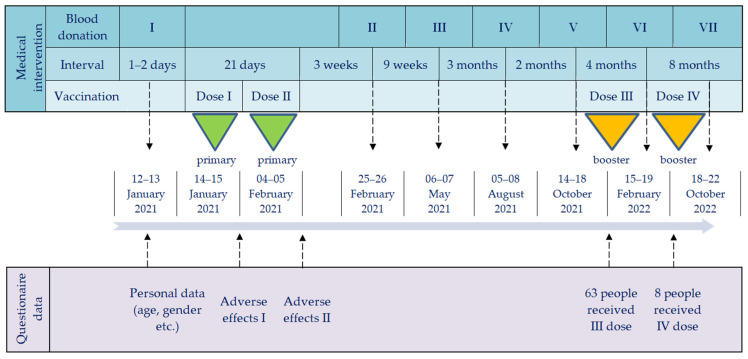
Timeline of vaccine administrations, blood donation checkpoints, and conducted questionnaires. Abbreviations: Roman numerals (I–VII) and arrows, subsequent blood and data collections; green triangles, primary vaccination; orange triangles, booster doses.

**Figure 2 vaccines-11-01578-f002:**
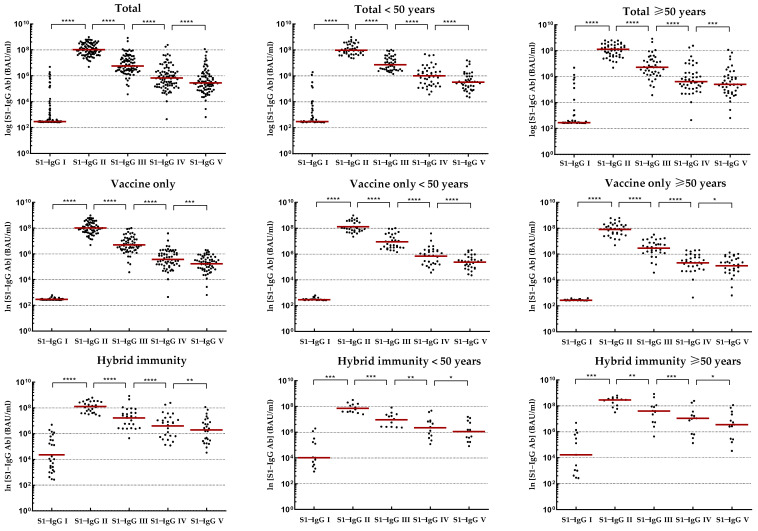
Kinetics of IgG-S1 antibodies in relevant study cohorts. Abbreviations: * *p*-Value ≤ 0.05; ** *p*-Value ≤ 0.01; *** *p*-Value ≤ 0.001; **** *p*-Value ≤ 0.0001. Results obtained for individual patients are represented by dots. Red lines indicate median values.

**Figure 3 vaccines-11-01578-f003:**
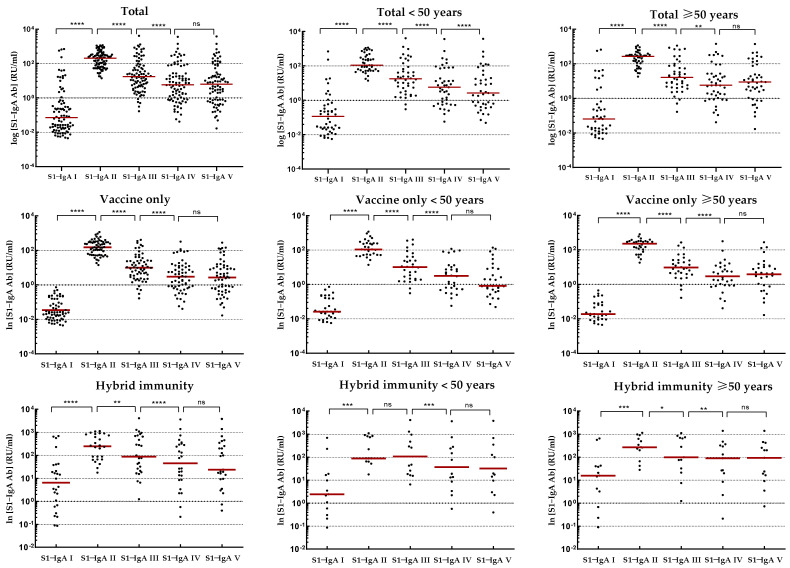
Kinetics of IgA-S1 antibodies in relevant study cohorts. Abbreviations: ^ns^ *p*-Value > 0.05; * *p*-Value ≤ 0.05; ** *p*-Value ≤ 0.01; *** *p*-Value ≤ 0.001; **** *p*-Value ≤ 0.0001; ns, not significant.. Results obtained for individual patients are represented by dots. Red lines indicate median values.

**Figure 4 vaccines-11-01578-f004:**
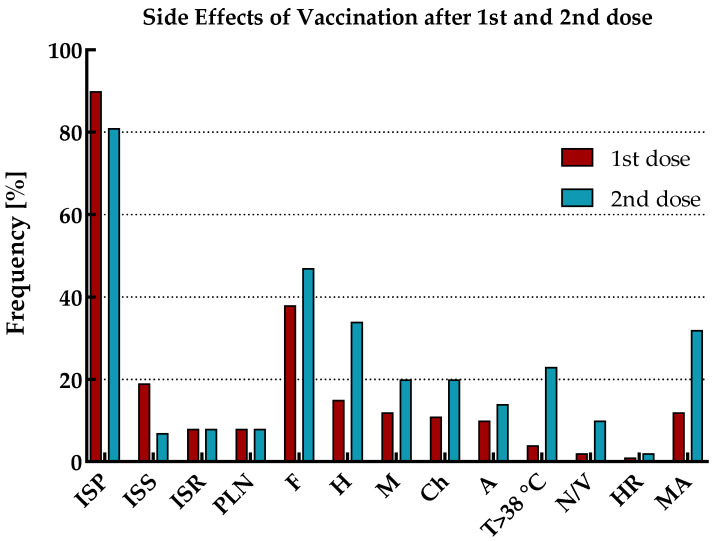
General prevalence of side effects of vaccination after first and second dose.

**Figure 5 vaccines-11-01578-f005:**
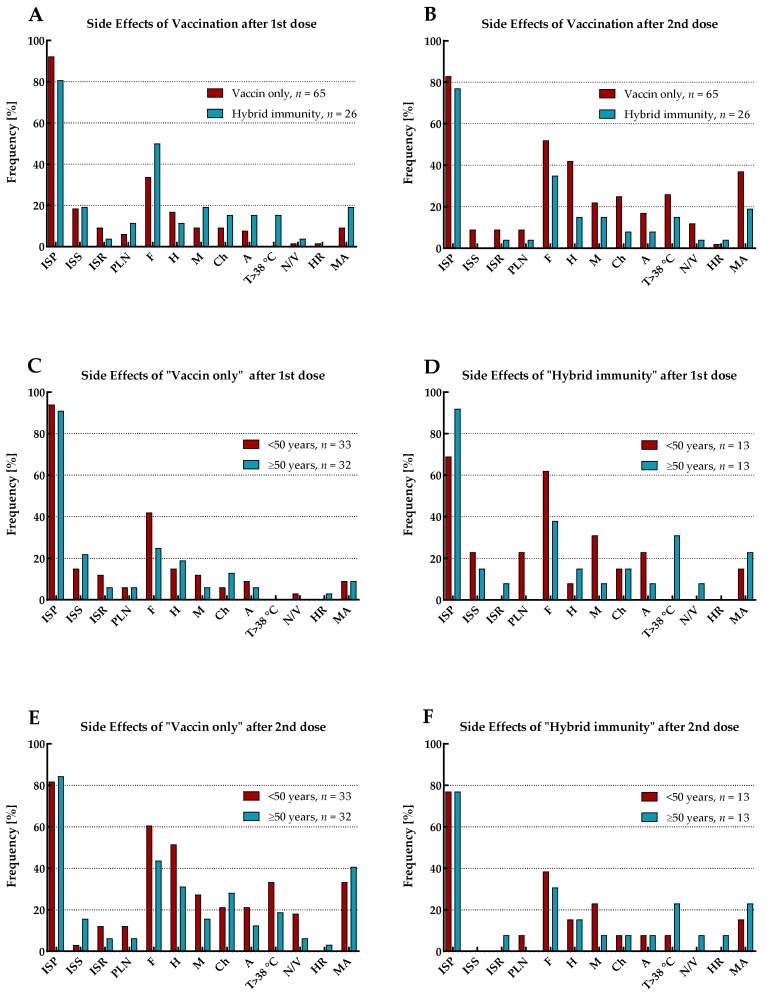
Prevalence of side effects of vaccination after the first and second dose by immune status and age range. Legend: (**A**) SEV after the first vaccine dose considering immune status. (**B**) SEV after the second vaccine dose considering immune status. (**C**) SEV after the first dose in participants of ‘vaccine only’ status considering age. (**D**) SEV after the first dose in participants of ‘hybrid’ immunity considering age. (**E**) SEV after the second dose in participants of ‘vaccine only’ status considering age. (**F**) SEV after the second dose in participants of ‘hybrid’ immunity considering age.

**Table 1 vaccines-11-01578-t001:** Characteristics of study participants.

Variables	No. (%) of Participants			*p*-Value ^a^
	Total	<50 Years	≥50 Years	
Gender: Female Male	72 (79%) 19 (21%)	37 (41%) 9 (10%)	35 (38%) 10 (11%)	0.8474 0.8514
Age: Median/Mean (IQR ^b^, min–max)	49.0/47.5 (17.0, 25–80)	39.5/37.9 (13.5, 25–49)	56.0/57.4 (7.0, 50–80)	<0.0001
Chronic disease ^c^: Hypertension Diabetes mellitus Chronic heart disease Chronic respiratory disease Autoimmune/autoinflammatory Cancer	30 (33.0%) 10 (11.0%) 2 (2.2%) 3 (3.3%) 2 (2.2%) 12 (13.2%) 1 (1.1%)	12 (26.1%) 3 (6.5%) 1 (2.2%) 1 (2.2%) 1 (2.2%) 6 (13.0%) 0 (0.0%)	18 (40.0%) 7 (15.6%) 1 (2.2%) 2 (4.4%) 1 (2.2%) 6 (13.3%) 1 (2.2%)	0.3158 0.2171 0.9877 0.5573 0.9877 0.9715 0.0163
BMI ^d^: Median (IQR) Underweight Optimal weight Overweight Obese I Obese II BMI Female Median (IQR) BMI Male Median (IQR)	24.72 (6.6) 6 (6.6%) 45 (49.4%) 27 (29.7%) 10 (11.0%) 3 (3.3%) 24.06 (6.9) 25.73 (3.1)	23.33 (5.4) 6 (13.0%) 25 (54.3%) 10 (21.7%) 3 (6.5%) 2 (4.3%) 22.84 (4.7) 25.96 (6.9)	25.66 (5.2) 0 (0.0) 20 (44.4%) 17 (37.8%) 7 (15.6%) 1 (2.2%) 25.95 (7.5) 25.38 (1.9)	0.0045 0.0339 0.5425 0.2696 0.2974 0.6353 0.0028 0.6607
Immunity status: Vaccine only Hybrid	65 (71%) 26 (28%)	33 (36%) 13 (14%)	32 (35%) 13 (14%)	0.9193 >0.9999

^a^ *p*-Value calculated using a Mann–Whitney U test or Chi-square test. ^b^ IQR, inter-quartile range. ^c^ A total of 23 participants reported chronic disease(s), of which 17, 5, and 1 were characterised by single, two, and three morbidities, respectively. ^d^ BMI, body mass index; the participants were divided into groups including underweight, optimal weight, overweight, obesity I, and obesity II with BMI < 18.49, 18.5–24.99, 25–29.99, 30–34.99 and ≥35, respectively.

**Table 2 vaccines-11-01578-t002:**
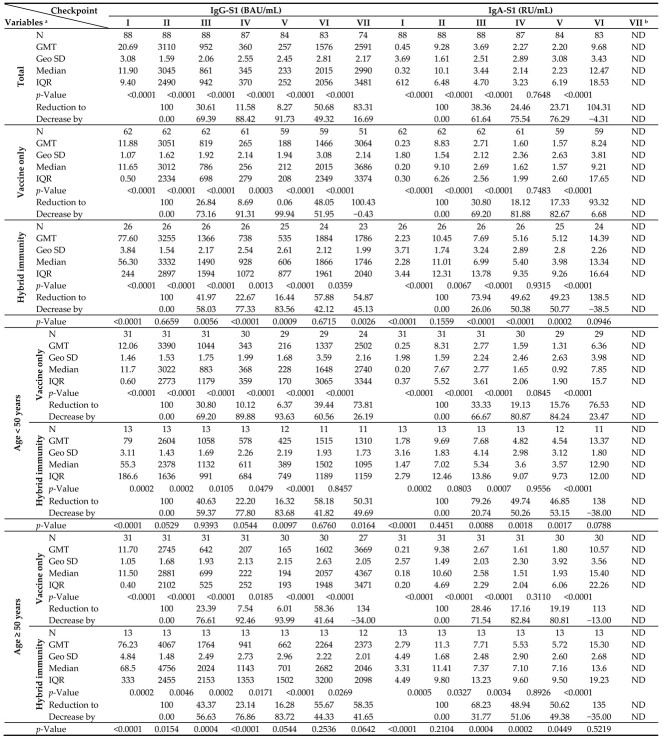
Kinetics of IgG-S1 and IgA-S1 antibodies at checkpoints for the total cohort and by immune status (‘vaccine-only’/’hybrid’ immunity), as well as by age (above/below 50 years).

^a^ Abbreviations: GMT, geometric mean titre; Geo SD, the range from the GMT divided by the geometric SD factor to the GMT multiplied by the geometric SD factor will contain about two thirds of the values if the data are sampled from a lognormal distribution; IQR, inter-quartile range; *p*-Value calculated using an exact two-tailed Wilcoxon matched-pairs signed rank test for an abnormal distribution for seroconversion and Mann–Whitney test between cohorts—‘Hybrid immunity’ and ‘Vaccine only’. ^b^ ND, not data.

**Table 3 vaccines-11-01578-t003:**
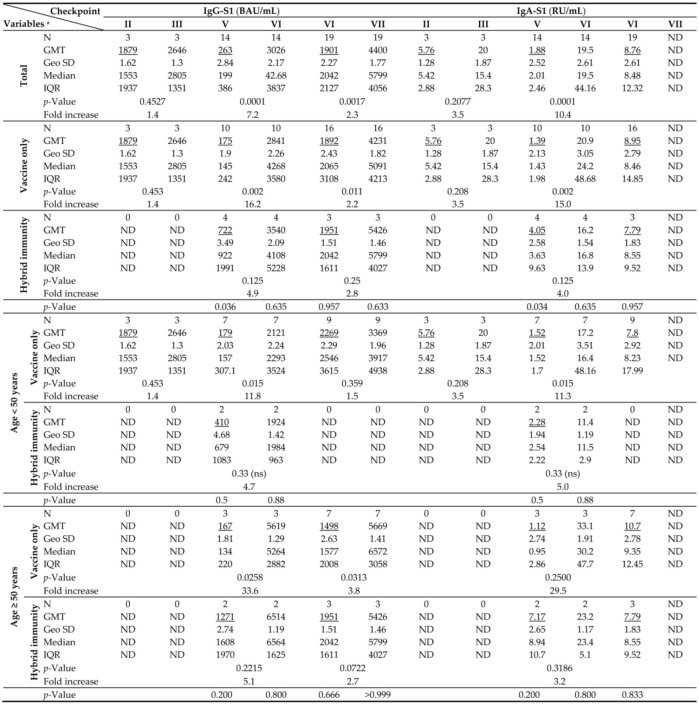
Breakthrough levels (underlined) and seroconversions of IgG-S1 and IgA-S1 levels at subsequent checkpoints for the total cohort, by immune status (‘vaccine only’/’hybrid’ immunity) as well as by age (over/under 50 years).

^a^ Abbreviations: GMT, geometric mean titre; Geo SD, the range from the GMT divided by the geometric SD factor to the GMT multiplied by the geometric SD factor will contain about two thirds of the values if the data are sampled from a lognormal distribution; IQR, inter-quartile range; *p*-Value calculated using an exact two-tailed Wilcoxon matched-pairs signed rank test for an abnormal distribution for seroconversion and Mann–Whitney test between cohorts—‘Hybrid immunity’ and ‘Vaccine only’. ^b^ ND, no data.

**Table 4 vaccines-11-01578-t004:** Correlations of anti-SARS-CoV-2 antibody levels between recovered and uninfected individuals and number of received vaccine dose(s).

Variables ^a^		No Infection					Infection Between the V and VI Donation					Infection after the VI Donation ^b^				
		IgG-S1 (BAU/mL)			IgA-S1 (RU/mL)		IgG-S1 (BAU/mL)			IgA-S1 (RU/mL)		IgG-S1 (BAU/mL)			IgA-S1 (RU/mL)	
		V	VI	VII	V	VI	V	VI	VII	V	VI	V	VI	VII	V	VI
**No booster**	N GMT Geo SD Median IQR	12 257 1.33 255 121	12 132 1.33 133 52	12 39.9 2.15 40.3 30.6	12 1.66 3.61 0.9 5.13	12 0.72 3.87 0.48 2.33	2 129 1.31 132 50	2 2869 2.36 3414 3700	2 1691 2.32 2001 2139	2 1.09 3.48 1.54 2.17	2 12.0 1.82 13.0 10.36	0 ND ND ND ND	0 ND ND ND ND	ND ND ND ND ND	0 ND ND ND ND	0 ND ND ND ND
**Booster I**	N GMT Geo SD Median IQR	39 221 2.25 207 194	39 1805 2.16 2017 1563	39 2455 2.16 2732 3312	39 2.65 2.72 2.66 4.67	39 12.9 2.86 14.5 22.41	10 258 2.47 284 392	10 2621 2.2 3252 3262	10 2328 1.58 2260 1800	10 1.62 2.0 1.48 2.47	10 18.9 2.81 21.2 44.16	14 288 1.61 253 157	14 2072 2.1 2481 1566	14 4825 1.68 6046 3413	14 2.05 2.64 1.67 4.31	14 8.61 2.05 8.43 9.88
**Booster II**	N GMT Geo SD Median IQR	4 203 7.16 334 1251	4 1059 5.41 2671 2797	4 1694 2.49 1778 3451	4 1.27 5.92 2.45 6.22	4 3.42 6.38 7.85 12.09	1 2593 1.0 2593 0.0	1 5752 1.0 5752 0.0	1 7003 1.0 7003 0.0	1 14.3 1.0 14.3 0.0	1 20.8 1.0 20.8 0.0	3 193 1.93 134 283	3 1840 3.47 1230 6740	3 5414 1.65 7099 5414	3 1.41 4.18 3.11 3.04	3 11.4 7.14 16.4 64.54

^a^ Abbreviations: GMT, geometric mean titre; Geo SD, the range from the GMT divided by the geometric SD factor to the GMT multiplied by the geometric SD factor will contain about two thirds of the values if the data are sampled from a lognormal distribution; IQR, inter-quartile range. ^b^ ND, no data.

**Table 5 vaccines-11-01578-t005:** Prevalence of side effects of vaccination with mean duration and reported medication admission in relevant study cohorts.

	**SEV after 1st Dose ^a^**		**ISP**	**ISS**	**ISR**	**PLN**	**F**	**H**	**M**	**Ch**	**A**	**T > 38 °C**	**N/V**	**HR**	**MA**	**Median**	**Mean**
**Total**	*n* = 91	No. of SEV	82	17	7	7	35	14	11	10	9	4	2	1	11	9.5	16.6
		%	90%	19%	8%	8%	38%	15%	12%	11%	10%	4%	2%	1%	12%		
		Time (days)	2.1	0.6	0.3	0.4	0.6	0.2	0.4	0.1	0.4	0.1	0.0	0.2	0.2	0.3	0.4
	<50 years *n* = 46	No. of SEV	41	8	4	5	22	6	8	4	6	0	1	0	5	5.5	8.8
		%	89%	17%	9%	11%	48%	13%	17%	9%	13%	0%	2%	0%	11%		
		Time (days)	1.9	0.7	0.5	0.6	0.9	0.2	0.6	0.1	0.6	0.0	0.0	0.0	0.2	0.6	0.5
	≥50 years *n* = 45	No. of SEV	41	9	3	2	13	8	3	6	3	4	1	1	6	0.1	0.2
		%	91%	20%	7%	4%	29%	18%	7%	13%	7%	9%	2%	2%	13%		
		Time (days)	2.3	0.4	0.1	0.1	0.4	0.2	0.1	0.1	0.2	0.1	0.0	0.5	0.2	0.1	0.4
**Vaccine only**	*n* = 65	No. of SEV	60	12	6	4	22	11	6	6	5	0	1	1	6	6.0	11.2
		%	92%	18%	9%	6%	34%	17%	9%	9%	8%	0%	2%	2%	9%		
		Time (days)	2.1	0.6	0.4	0.3	0.6	0.2	0.2	0.1	0.3	0.0	0.0	0.3	0.2	0.3	0.4
	<50 years *n* = 33	No. of SEV	31	5	4	2	14	5	4	2	3	0	1	0	3	3.5	5.9
		%	94%	15%	12%	6%	42%	15%	12%	6%	9%	0%	3%	0%	9%		
		Time (days)	2.1	0.7	0.7	0.5	0.8	0.2	0.3	0.0	0.3	0.0	0.0	0.0	0.1	0.3	0.5
	≥50 years *n* = 32	No. of SEV	29	7	2	2	8	6	2	4	2	0	0	1	3	2.0	5.3
		%	91%	22%	6%	6%	25%	19%	6%	13%	6%	0%	0%	3%	9%		
		Time (days)	2.2	0.5	0.1	0.1	0.3	0.1	0.1	0.1	0.3	0.0	0.0	0.7	0.2	0.1	0.4
**Hybrid immunity**	*n* = 26	No. of SEV	21	5	1	3	13	3	5	4	4	4	1	0	5	2.0	5.3
		%	81%	19%	4%	12%	50%	12%	19%	15%	15%	15%	4%	0%	19%		
		Time (days)	2.0	0.6	0.0	0.4	0.8	0.2	0.7	0.1	0.7	0.2	0.1	0.0	0.3	0.1	0.4
	<50 years *n* = 13	No. of SEV	9	3	0	3	8	1	4	2	3	0	0	0	2	2.5	2.8
		%	69%	23%	0%	23%	62%	8%	31%	15%	23%	0%	0%	0%	15%		
		Time (days)	1.4	0.9	0.0	0.8	1.2	0.1	1.4	0.1	1.4	0.0	0.0	0.0	0.5	0.5	0.6
	≥50 years *n* = 13	No. of SEV	12	2	1	0	5	2	1	2	1	4	1	0	3	1.5	2.6
		%	92%	15%	8%	0%	38%	15%	8%	15%	8%	31%	8%	0%	23%		
		Time (days)	2.5	0.3	0.0	0.0	0.5	0.2	0.1	0.1	0.1	0.5	0.2	0.0	0.2	0.1	0.4
**SEV after 2nd dose ^a^**			**ISP**	**ISS**	**ISR**	**PLN**	**F**	**H**	**M**	**Ch**	**A**	**T > 38 °C**	**N/V**	**HR**	**MA**	**Median**	**Mean**
**Total**	*n* = 91	No. of SEV	74	6	7	7	43	31	18	18	13	21	9	2	29	15.5	20.8
		%	81%	7%	8%	8%	47%	34%	20%	20%	14%	23%	10%	2%	32%		
		Time (days)	1.8	0.2	0.2	0.3	1.0	0.7	0.5	0.2	0.7	0.2	0.1	0.1	0.9	0.3	0.5
	<50 years *n* = 46	No. of SEV	37	1	4	5	25	19	12	8	8	12	6	0	13	8	11.4
		%	80%	2%	9%	11%	54%	41%	26%	17%	17%	26%	13%	0%	28%		
		Time (days)	1.9	0.0	0.2	0.3	1.3	1.0	0.9	0.2	0.8	0.3	0.2	0.0	0.9	0.3	0.6
	≥50 years *n* = 45	No. of SEV	37	5	3	2	18	12	6	10	5	9	3	2	16	5.5	9.3
		%	82%	11%	7%	4%	40%	27%	13%	22%	11%	20%	7%	4%	36%		
		Time (days)	1.8	0.3	0.2	0.3	0.6	0.3	0.1	0.3	0.5	0.2	0.1	0.2	0.8	0.3	0.4
**Vaccine only**	*n* = 65	No. of SEV	54	6	6	6	34	27	14	16	11	17	8	1	24	12.5	16.7
		%	83%	9%	9%	9%	52%	42%	22%	25%	17%	26%	12%	2%	37%		
		Time (days)	1.9	0.2	0.2	0.4	1.1	0.8	0.5	0.3	0.8	0.3	0.2	0.1	1.0	0.4	0.6
	<50 years *n* = 33	No. of SEV	27	1	4	4	20	17	9	7	7	11	6	0	11	7.0	9.4
		%	82%	3%	12%	12%	61%	52%	27%	21%	21%	33%	18%	0%	33%		
		Time (days)	1.8	0.1	0.2	0.4	1.4	1.1	0.9	0.3	0.8	0.3	0.2	0.0	0.9	0.4	0.6
	≥50 years *n* = 32	No. of SEV	27	5	2	2	14	10	5	9	4	6	2	1	13	5.0	7.3
		%	84%	16%	6%	6%	44%	31%	16%	28%	13%	19%	6%	3%	41%		
		Time (days)	2.0	0.4	0.1	0.5	0.7	0.4	0.2	0.3	0.7	0.2	0.1	0.2	1.1	0.4	0.5
**Hybrid immunity**	*n* = 26	No. of SEV	20	0	1	1	9	4	4	2	2	4	1	1	5	2.0	4.1
		%	77%	0%	4%	4%	35%	15%	15%	8%	8%	15%	4%	4%	19%		
		Time (days)	1.7	0.0	0.2	0.0	0.8	0.5	0.6	0.1	0.4	0.1	0.1	0.1	0.6	0.1	0.4
	<50 years *n* = 13	No. of SEV	10	0	0	1	5	2	3	1	1	1	0	0	2	1.0	2.0
		%	77%	0%	0%	8%	38%	15%	23%	8%	8%	8%	0%	0%	15%		
		Time (days)	2.1	0.0	0.0	0.1	1.2	0.8	1.2	0.0	0.8	0.0	0.0	0.0	0.9	0.1	0.5
	≥50 years *n* = 13	No. of SEV	10	0	1	0	4	2	1	1	1	3	1	1	3	1.0	2.1
		%	77%	0%	8%	0%	31%	15%	8%	8%	8%	23%	8%	8%	23%		
		Time (days)	1.3	0.0	0.3	0.0	0.4	0.1	0.1	0.1	0.1	0.2	0.2	0.1	0.2	0.1	0.2

^a^ Abbreviations: SEV—side effects of vaccination; ISP—pain; ISS—swelling; ISR—redness; PLN—pain of lymph nodes; F—fatigue; H—headache; M—myalgia; Ch—chills; A—arthralgia; T > 38 °C-fever; N/V—nausea or vomiting; HR—hypersensitivity reactions; MA—medications.

## Data Availability

Raw data will be available upon suitable request to the corresponding author.

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
