# Peer review of "Twenty-Month Monitoring of Humoral Immune Response to BNT162b2 Vaccine: Antibody Kinetics, Breakthrough Infections, and Adverse Effects"

_vaccines, 2023, doi:10.3390/vaccines11101578_

Round 1

Reviewer 1 Report

The objectives of the current study were to: assess the humoral immune response to BNT162b2 vaccine and SARS-CoV-2 infection; demonstrate the persistence of anti-SARS-CoV-2 antibodies over time in relation to the number of received vaccine doses and the course of infection; and determine the adverse effects after primary vaccine doses. The manuscript is well organized and written and it discuss an interesting topic with a field application. The study design and data collections are well planned and prepared. The main limitation of this study is the small sample size and this was discussed in the limitation at the end of discussion section. In general, the manuscript can be accepted after few amendments listed below:

L21: add "the " before "adverse effects"

L35-36: try to add other keywords that were not mentioned in the title

L46-48: show that this number is "worldwide"

L155: add "the" before "third blood donations"

L326-337: the symptoms should be written in full word.

L442: general recommendations should be added at the end of conclusions section.

Author Response

Rev 1

The objectives of the current study were to: assess the humoral immune response to BNT162b2 vaccine and SARS-CoV-2 infection; demonstrate the persistence of anti-SARS-CoV-2 antibodies over time in relation to the number of received vaccine doses and the course of infection; and determine the adverse effects after primary vaccine doses. The manuscript is well organized and written and it discuss an interesting topic with a field application. The study design and data collections are well planned and prepared. The main limitation of this study is the small sample size and this was discussed in the limitation at the end of discussion section.

We would like to thank the Reviewer for his/her positive comments and useful suggestions to improve the manuscript.

In general, the manuscript can be accepted after few amendments listed below:

L21: add "the " before "adverse effects"

It has been done.

L35-36: try to add other keywords that were not mentioned in the title

It has been corrected.

L46-48: show that this number is "worldwide"

It has been done.

L155: add "the" before "third blood donations"

It has been done.

L326-337: the symptoms should be written in full word.

It has been corrected.

L442: general recommendations should be added at the end of conclusions section.

It has been corrected.

Reviewer 2 Report

This manuscript described IgG and IgA Kinetics after BNT162b2 vaccination or breakthrough infections, which can contribute to the understanding of the change tendency of SARS-CoV-2 antibody. However, the manuscript must be revised before publishing.

1.  How the participants were recruited? How about the representativeness? How the sample size was evaluated? These details should be addressed.

2. The definition of “hybrid” is needed.  

3. The result about IgG-NCP is missing.

4. When breakthrough infections occur? It’s hard to understand “To track trends in antibody levels after primary vaccine doses in intact systems (with vaccine-induced immunity only), three individuals who contracted SARS-CoV-2 infection between the second and third blood donations were excluded from the analysis.”

5. The contents of Table 2 and Fig2 overlapped.

6. Table 2: what the “Percentage” and “Decrease” refers to?

7. What matched-pairs do p-Values refers to?

8. Many expressions are hard to understand. For example:  “Joining the study, 26 individuals possessed a recovered status (positive results for IgG-NCP in the first blood donation) with higher antibody levels for IgG-S1 and IgA-S1 classes (7- and 10-fold, respectively) than in uninfected subjects. Please revise such sentences.

9. What’s the evidence in this manuscript for conclusion “Vaccination is highly effective in preventing the most severe outcomes of COVID-19 ”?

10. How to guarantee all breakthrough infections were identified? 

Many sentences are hard to understand.

Author Response

This manuscript described IgG and IgA Kinetics after BNT162b2 vaccination or breakthrough infections, which can contribute to the understanding of the change tendency of SARS-CoV-2 antibody. However, the manuscript must be revised before publishing.

We would like to thank the Reviewer for his/her positive comments and useful suggestions to improve the manuscript.

  1. How the participants were recruited? How about the representativeness? How the sample size was evaluated? These details should be addressed.

Study participants were recruited from among employees of the National Institute of Medicine who were willing to monitor the post-vaccination response and possible SARS-CoV-2 infections. In addition, study participants had to participate in at least the first 3 checkpoints. Prior to the start of the study, neither the patients nor the researchers knew which study groups they would be classified into and what the final size of the groups would be. The groups were formed and compared based on the analysis of the post-study data.

The main criterion for subdivision was previous confirmed SARS-CoV-2 infection prior to baseline vaccination. The control group consisted of individuals without earlier infection.

The eligible number of subjects (n=91) is representative of the population of individuals >18y (30 millions), assuming confidence in the results obtained α = 0.95 and a maximum error of the result of 10%. At this confidence level, our results estimate the true results in the population with a maximum error of 10%.

In the Materials and Methods section, under 2.1. Study design, data collection and cohort characteristics, the information presented above has been added. The Discussion section has also been changed (lines 430-431).

  1. The definition of “hybrid” is needed.

In the study, hybrid immunity was defined as immune protection in individuals who received at least two doses of COVID-19 vaccine and experienced at least one confirmed SARS-CoV-2 infection before vaccination. Infections after baseline vaccination were classified as breakthrough infections.

Explanations of "hybrid resistance" are included in Introduction (lines: 67-69) and Results (lines 265-270).

  1. The result about IgG-NCP is missing.

The results for IgG-NCP have been added in Table S1. We introduced the information about Table S1 in the text.

  1. When breakthrough infections occur? It’s hard to understand “To track trends in antibody levels after primary vaccine doses in intact systems (with vaccine-induced immunity only), three individuals who contracted SARS-CoV-2 infection between the second and third blood donations were excluded from the analysis.”

We considered that the breakthrough infections occurred 2-3 weeks after the second dose of the vaccine. Thank you for pointing out our mistake; we have corrected the indicated sentence.

  1. The contents of Table 2 and Fig2 overlapped.

Yes, we agree with the Reviewer's opinion. However, in order to make it clearer for the average reader to follow the decline in antibody levels and the range of scatter in the results, we have added Figure 2 and Figure 3.

  1. Table 2: what the “Percentage” and “Decrease” refers to?

In Table 2, "percent" refers to the maximum antibody level obtained after primary vaccination, which was always at the 2nd intake. At subsequent checkpoints, it is the percentage of the maximum antibody level obtained (we changed to "reduction to"). "Decrease" is the level of decrease in antibody titter at subsequent checkpoints relative to the highest antibody level obtained in the group also expressed as a percentage (we changed to "decrease by").

We have modified the Table 2.

  1. What matched-pairs do p-Values refers to?

The p-Values placed between individual donations refer to their comparisons. An explanation is provided below the tables (p-value calculated using Wilcoxon's two-sided exact rank-sum test comparisons between levels Ab blood donations). The p-Values placed directly below the donations refer to comparisons between the cohorts - "Hybrid immunity" and "Vaccine only". Explanatory notes appear below the tables.

We have corrected the explanations under the tables.

  1. Many expressions are hard to understand. For example:  “Joining the study, 26 individuals possessed a recovered status (positive results for IgG-NCP in the first blood donation) with higher antibody levels for IgG-S1 and IgA-S1 classes (7- and 10-fold, respectively) than in uninfected subjects. Please revise such sentences.

It has been corrected.

  1. What’s the evidence in this manuscript for conclusion “Vaccination is highly effective in preventing the most severe outcomes of COVID-19 ”?

We agree with the Reviewer's suggestion. Conclusions have been modified.

  1. How to guarantee all breakthrough infections were identified? 

We are confident that all breakthrough infections were identified due to the parallel monitoring of the study group also for SARS-CoV-2 infections. The company provided all workers with the opportunity for free and unrestricted COVID-19 diagnosis in our laboratory. Any suspected infection, even a cold, was verified by RT-qPCR or rapid antigen test.

Reviewer 3 Report

What have you learnt new in BNT162b2 vaccine immunogenecity? What are the implications of your study results for public health/population vaccination coverage? 

Author Response

What have you learnt new in BNT162b2 vaccine immunogenecity?

To our knowledge, we have shown for the first time an almost complete decrease in both IgG-S1 and IgA-S1 was observed twenty months after primary vaccination in those who remained uninfected and did not receive any booster dose. We provided this information in the results (lines 296-298) and discussion (lines 390-391) of the manuscript.

What are the implications of your study results for public health/population vaccination coverage? 

As we wrote in the manuscript, it seems that full vaccination of convalescents is entirely justified, and that vaccination of people without previous COVID-19 disease with booster doses is even necessary (lines 380-383). In addition, since vaccination primarily prevents the severe effects of COVID-19, it seems that it should be performed regardless of the patient's characteristics, unless there are obvious obstacles to doing otherwise (lines 389-391). We also addressed the safety of COVID-19 vaccination in the manuscript. Redness and swelling at the injection site were the most common adverse reactions. Such information may be useful for those concerned about side effects.

Reviewer 4 Report

The manuscript by Walory and colleagues covers topic of SARS-CoV-2 prevention and humoral immune response to vaccination and natural infection. The manuscript contains solid data regarding S1-specific antibodies of two subclasses in various types of donors. Authors also studied antibody levels in patients with breakthrough infections. Manuscript could be of interest for specialists in vaccine development.

The major issue of the manuscript is that humoral immune response is analyzed only by the means of ELISA (with Wuhan-derived S1 protein most likely). Levels of antibodies specific for other VOCs including Delta and Omicron-like could be quite different. The quantity of antibodies measured by ELISA only slightly correlates with the live virus neutralization (especially neutralization of modern VOCs).

The discussion section of the manuscript lacks a comparison of the presented data with the published ones. There are hundreds of papers devoted to SARS-CoV-2 specific humoral immune response monitoring, so authors should highlight novelty of their research.

Author Response

The manuscript by Walory and colleagues covers topic of SARS-CoV-2 prevention and humoral immune response to vaccination and natural infection. The manuscript contains solid data regarding S1-specific antibodies of two subclasses in various types of donors. Authors also studied antibody levels in patients with breakthrough infections. Manuscript could be of interest for specialists in vaccine development.

We would like to thank the Reviewer for his/her positive comments.

The major issue of the manuscript is that humoral immune response is analyzed only by the means of ELISA (with Wuhan-derived S1 protein most likely). Levels of antibodies specific for other VOCs including Delta and Omicron-like could be quite different. The quantity of antibodies measured by ELISA only slightly correlates with the live virus neutralization (especially neutralization of modern VOCs).

The main idea of our study was to determine the kinetics of post-vaccination antibodies over time. We monitored the humoral response after vaccination with the BNT162b2 vaccine, which contains a nucleoside-modified mRNA encoding the S protein specific for the Wuhan-Hu-1 strain. Therefore, it was necessary to determine the anti-S1 antibodies from the 'vaccine strain'. 

The discussion section of the manuscript lacks a comparison of the presented data with the published ones. There are hundreds of papers devoted to SARS-CoV-2 specific humoral immune response monitoring, so authors should highlight novelty of their research.

We have updated (for September 2023) the time of access to reference items that include websites. The references used in the manuscript mainly concern articles from 2021-2023, which were most relevant to the discussed by us issues. As the Reviewer himself noted, every month there are many new publications covering the aspect we are dealing with, so it is not possible to include all of them.

Round 2

Reviewer 2 Report

What I concerned has been revised. I recommend accepting this manuscript. 

Reviewer 4 Report

Authors corrected discussion, paper could be published.